# No evidence of host-specific egg mimicry in Asian koels

Mominul Islam Nahid [1,2]*, Frode Fossøy[1,3], Bård G. Stokke[1,3], Virginia Abernathy[4], Sajeda Begum[2], Naomi E. Langmore[4], Eivin Røskaft[1], Peter S. Ranke[5]

1 Department of Biology, Norwegian University of Science and Technology (NTNU), Trondheim, Norway,
2 Jahangirnagar University, Savar, Dhaka, Bangladesh, 3 Norwegian Institute for Nature Research (NINA), Trondheim, Norway, 4 Research School of Biology, Australian National University, Canberra, Australian Capital Territory, Australia, 5 Centre for Biodiversity Dynamics (CBD), Department of Biology, Norwegian University of Science and Technology (NTNU), Trondheim, Norway

* nahid_1511@yahoo.com

## Abstract

Avian brood parasitism is costly for the host, in many cases leading to the evolution of defenses like discrimination of parasitic eggs. The parasite, in turn, may evolve mimetic eggs as a counter-adaptation to host egg rejection. Some generalist parasites have evolved host-specific races (gentes) that may mimic the eggs of their main hosts, while others have evolved 'jack-of-all-trades' egg phenotypes that mimic key features of the eggs of several different host species. The Asian koel (*Eudynamys scolopaceus*) is a widely distributed generalist brood parasite that exploits a wide range of host species. Based on human vision, previous studies have described Asian koel eggs as resembling those of its main host, the house crow (*Corvus splendens*). Using measurements of egg length and breadth, digital image analysis, reflectance spectrophotometry and avian visual modelling, we examined Asian koel egg variation and potential mimicry in egg size and shape, and eggshell pattern and color in three sympatrically occurring host species in Bangladesh: the common myna (*Acridotheres tristis*), house crow, and long-tailed shrike (*Lanius schach*). We found some differences among Asian koel eggs laid in different host nests: a) Asian koel eggs in long-tailed shrike nests were larger than those laid in common myna and house crow nests, and b) Asian koel eggs in house crow nests were less elongated than those in common myna nests. However, these changes in Asian koel egg volume and shape were in the opposite direction with respect to their corresponding host egg characteristics. Thus, our study found no evidence for Asian koel host-specific egg mimicry in three sympatrically occurring host species.

## Introduction

Avian interspecific brood parasites lay their eggs in the nests of other species, the hosts, and trick them into raising their young, thereby avoiding the cost of parental care [1,2]. Successful brood parasitism is, in general, costly for the host species as it usually severely reduces or

Data Availability Statement: All relevant data are within the manuscript and its Supporting information files.

Funding: The study was funded by grants through 'Quota Scheme' at Norwegian University of Science

and Technology (NTNU), from NTNU, Department of Biology to Professor Eivin Røskaft, an Australian National University travel grant to Virginia Abernathy and Research Council of Norway (Centre of Excellence Grant 223257) to Peter S. Ranke. The funders had no role in study design, data collection and analysis, decision to publish, or preparation of the manuscript.

**Competing interests:** The authors have declared that no competing interests exist.

completely obliterates reproductive fitness of the host [2–5]. The high costs of parasitism drive the evolution of anti-parasite traits such as discrimination and rejection of parasitic eggs, i.e. identification and removal of the parasitic eggs from the nest [6,7]. Hosts use egg color [8–11], pattern [10,12–14], UV reflectance [e.g. 15–17], size [18–20, but see 21] and shape [22–25, but see 26] to identify and reject parasitic eggs in the clutch [for review, see 27, 28].

As a counter-adaptation to egg rejection, the parasite may evolve eggs that mimic host eggs in order to make it more difficult for hosts to recognize the parasitic egg [6,7, for review, see 27–30]. Although egg mimicry does not always evolve, potentially due to short time span of the host-parasite coevolutionary arms-race [30,31], host recognition abilities [9,31], metapopulation structures [32–34], exploitation of multiple hosts with different egg phenotypes [35–37] and possibly also climate variables [38], the evolution of egg mimicry is closely related to the evolution of host defenses (i.e., egg rejection) [30]. Furthermore, the evolution of egg mimicry may be more difficult to evolve for generalist than for specialist brood parasites, because the former have broader geographical distributions and exploit a wider variety of host species that lay eggs with different colors and patterns [39]. Some generalist brood parasites overcome this difficulty by evolving host-specific races, known as *gentes*, where each gens mimics the eggs of a particular host species [30,40,41]. Some generalist parasites may evolve mimicry of one of their main hosts, which later allows exploitation of new host species with similar egg phenotypes [42]. In other generalist parasites, host races may be absent, and they instead rely on a jack-of-all-trades mimicry, laying intermediate eggs relatively similar to all host eggs [46]. This strategy could only work if all of the hosts utilized lay eggs that are rather similar in appearance [43]. Even with the existence of host-specific races, the eggs of sympatric gentes may be similar in some egg traits [35–37], appearing to mimic the eggs of a range of hosts, even though parasitic females are mostly host-specific during egg laying [44,45]. However, some generalists show no or poor host egg mimicry (e.g. cowbirds, *Molothrus* sp.), because many of their hosts apparently lack the ability to reject foreign eggs [34,46,47]. Sometimes, parasite egg mimicry is not visible to the human eye due to differences between the human and avian visual systems [48,49]. Spectrophotometry has revealed that the pallid cuckoo (*Cacomantis pallidus*) lays mimetic, host-specific eggs that are not possible to detect by the human eye [50]. In contrast, the great spotted cuckoo (*Clamator glandarius*) was, as assessed by the human eye, assumed to lay mimetic eggs of one its main hosts, the magpie (*Pica pica*), but further examination by spectrophotometry revealed that there is no mimicry of the great spotted cuckoo eggs towards the host [51]. Some brood parasites lay cryptic rather than mimetic eggs when parasitizing hosts with dimly-lit nests [52–54]. In low-light nest environments, the cryptic egg blends in with the nest lining, making it difficult for the host and other parasites to see the parasite eggs in the nest [52–54]. Laying cryptic eggs may be advantageous for generalist brood parasites as it requires less host specialization. Evolution of cryptic eggs may be due to competing cuckoos rather than host rejection [27], as multiple parasitism within the same host nest can select for cryptic cuckoo eggs [52]. This is because, when cuckoos parasitize a nest, they typically remove one host egg whilst laying their own. Thus, when multiple parasitism occurs, the cuckoo egg that is laid first is at risk of being removed by the second-to-arrive cuckoo. However, if the cuckoo egg is cryptic within the nest, the second-to-arrive cuckoo is more likely to remove a host egg [52–54].

The Asian koel (*Eudynamys scolopaceus*) is a non-evicting brood parasite, meaning that the koel chick often coexists with the host chicks [55,56]. However, in many cases, the parasitic chick outcompetes the host chicks and significantly reduces the host's reproductive success [57]. The Asian koel has a wide distribution throughout Asia [55,56,58] which was documented 4000 years ago in ancient Vedic writings from India [1,59]. A total of seven subspecies of the Asian koel have been identified, with the nominate subspecies *E. s. scolopaceus* found in

south Asia [60]. It is a generalist brood parasite that regularly utilizes at least 16 host species differing greatly in body size, nest type, diet, habitat and distribution [55,56,61]. However, there is no information about whether individual parasitic females are host-specific, i.e. that they lay their eggs only (or predominantly) in nests of a single host species. The Asian koel lays eggs with a grey-bluish to greenish ground color, which are partly covered with numerous brown or black spots [55–57]. Previous studies, based on human vision, have described the Asian koel eggs as mimetic of the eggs of its oldest host, the house crow (*Corvus splendens*) [56–58,62–66]. Closely related species, such as the Pacific koel (*Eudynamis orientalis*) and the long-tailed cuckoo (*Urodynamis taitensis*), lay eggs with very different appearances from the Asian koel [42,67]. However, it should be noted that the egg morphology of the Pacific koel may have evolved due to an arms race with their main hosts in Australia, and thus may not reflect the ancestral state of egg morphology in Asian koels [42]. Hence, although the ancestral state of egg appearance in these species is unknown, egg morphology has probably evolved as a result of co-evolutionary interactions with their hosts.

In this study, we examine how Asian koel eggs vary in size, shape, spotting pattern and color, compared to specific host species, based on avian visual systems. We selected three regularly used host species within a single area in Bangladesh: including the oldest known host, the house crow, the most recent known host, long-tailed shrike (*Lanius schach*) and the common myna (*Acridotheres tristis*) [57,59,68]. A previous study investigating rejection of blue and brown model eggs found that house crows rejected 9.1%, common mynas rejected 0%, and long-tailed shrikes rejected 75% of such eggs [69]. To examine Asian koel egg mimicry, we explore the following questions: 1) is there variation in egg phenotypes among Asian koels laying eggs in different host nests? and 2) is there evidence that Asian koels mimic the eggs of their selected hosts?

## Materials and methods

### Ethics statement

All research and data collection were completed according to ethical laws of Bangladesh, and there was no injury to any birds in this study. The study was conducted on the Jahangirnagar University campus with the permission and monitoring of the Department of Zoology. The study was specifically approved by the review board of the wildlife and conservation biology branch, Department of Zoology after the evaluation of ethical laws of Bangladesh. The study site is an unprotected area, and all the study species are in the category of non-CITES, non-protected list of Bangladesh and least concern in the country's Red list.

### Study area

Fieldwork was carried out on the Jahangirnagar University campus (23˚52´ N, 90˚16´ E) including a teacher housing area, Arunapolli (23˚52´ N, 90˚17´ E), from 2008–2013 and 2015–2017. The study area is about 280 hectares and consists of diverse habitats including woodlands, grasslands, wetlands, cultivated lands and human settlements which make the campus area a fragmented habitat [70,71].

### Parasite and host community

The Asian koel is the most common avian brood parasite in the study area [68], parasitizing three species (overall parasitism rates for all years in parentheses): the long-tailed shrike (55.6%, n = 126), the common myna (33.6%, n = 271) and the house crow (16.4%, n = 165) (Nahid et al. submitted, see also [57]). Multiple parasitism by the Asian koel is common in all

these host species [68]. The house crow is the first mentioned host species of the Asian koel in the literature, documented around 375 A.D. in Sanskrit literature, and it is currently one of the most regularly exploited hosts across Asia [57,59,63,66], though the parasitism rate of this species is lower than the two other hosts at our study site [68]. The house crow builds open, shallow-cup nests and lays 3–6 eggs with bluish-green ground color and black or brown blotches [58,66,72]. However, a recent study confirmed that the house crow sometimes lays immaculate blue eggs [73]. In Bangladesh the common myna is also a regularly used host species and has become one of the main hosts of the Asian koel in our study area [57,68]. The common myna builds a variety of nest types that usually result in low-lit environments, including nesting in tree holes, inside buildings or roofs, or in the small space between joining palm leaves of coconut (*Cocos nucifera*), fishtail palm (*Caryota urens*) and fountain palm (*Livistona chinensis*) trees. Common mynas often also reconstruct old pied myna (*Gracupica contra*) or house crow nests. The eggs are immaculate blue and the clutch size is 4–5 eggs [58,74]. The long-tailed shrike is the most recent host species in Bangladesh [57,68]. This host builds small, open-cup shaped nests and lays 4–6 eggs with a pinkish-cream ground color and greyish and reddish spots on the blunt end [57,58]. Typical egg appearance of the Asian koel and its three corresponding host species is shown in Fig 1.

## General methodology

Measurements of host egg volume, shape, spotting pattern and color belonging to the same nest were included with nest identity as a random intercept, to account for non-independence of eggs within a clutch. All koel eggs were treated as independent measurements, even if they came from the same host nest because multiple parasitism was common (mean ± SD number of koel eggs in house crow nests: 1.33 ± 0.48, n = 27; common myna nests: 2.55 ± 1.60, n = 89; and long-tailed shrike nests: 2.05 ± 1.00, n = 66, unpublished results). Asian koel eggs with different sizes and ground color are often found in the same nest and therefore, we regard it as highly likely that more than one parasitic female laid eggs in a single host nest at our study site.

**Egg size and shape.** Egg length and breadth (mm) were measured using a digital calliper. Egg volume was estimated using the following formula: $V = (0.6057–0.0018B)LB^2$, where, $V =$ volume (mm$^3$), $B =$ breadth (mm) and $L =$ length (mm) [75]. Egg shape was calculated as the ratio $L/B$ [76].

**Egg pattern.** To measure egg pattern, we used digital photos of eggs. In 2015 we used a Canon EOS Kiss X5 camera with a 100 mm f/2.8 macro lens to photograph each egg placed on a 16% grey standard Kodak card. In 2016 and 2017 we used a Canon EOS 7D camera with a 100 mm f/2.8 macro lens to photograph the eggs placed on an 18% grey standard Kodak card. A tripod was used to retain a constant distance from the egg to the lens and to stabilize the camera. Common myna eggs were not included in this analysis as they are immaculate i.e. lack pattern.

All egg photos were analyzed in ImageJ [77] using the multispectral image calibration and analysis toolbox [78]. The toolbox accomplishes image calibration, confirming image linearization that regulates the lighting changes during image processing. To perform pattern analysis, standard band-pass methods on the camera's green reflectance channel was used as this is the closest approximation of bird double cone peak sensitivities [10,42]. Pattern energy spectrums for each egg were calculated at different spatial scales ranging from 2–512 pixels and the photos were scaled to 50 pixels/millimeter.

We performed granularity analysis based on fast Fourier band-pass filtering following the updated methods of Stoddard and Stevens [79], published by Troscianko and Stevens [78]. The granularity analysis produced several variables as summary statistics, and we used four

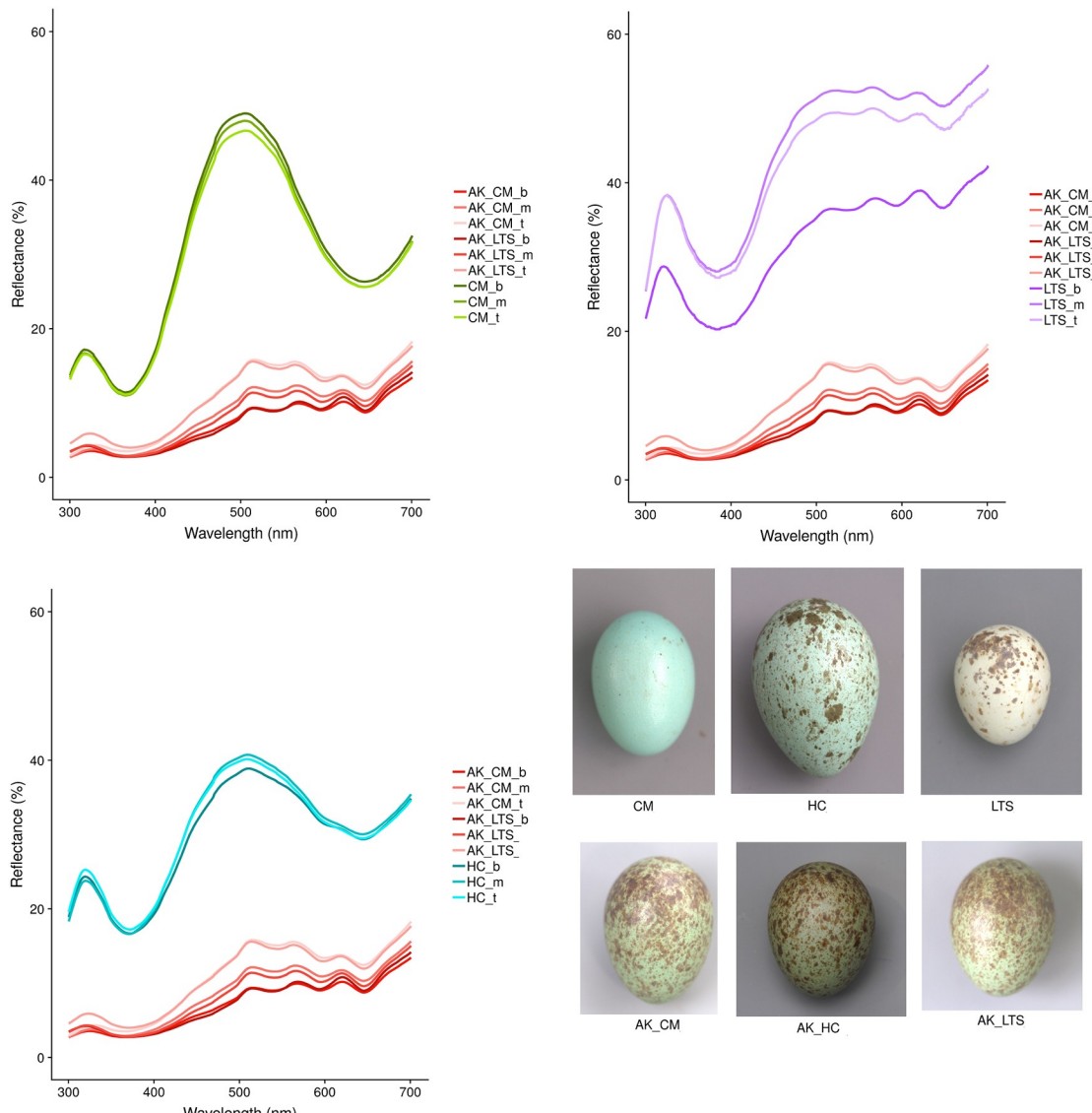

**Fig 1. Average spectral reflectance of three host species and Asian koel eggs laid in nests of two hosts (common myna and long-tailed shrike; note that house crow was not included due to lack of spectral reflectance data).** We include photographs of koel and host eggs from each species. All photos taken by V.E. Abernathy and M.I. Nahid. CM = common myna (n = 29 eggs from 11 nests), HC = house crow (n = 8 eggs from 3 nests), LTS = long-tailed shrike (n = 26 eggs from 7 nests), AK_CM = Asian koel in common myna nest (n = 5 eggs from 4 nests), AK_HC = Asian koel in house crow nest (not included) and AK_LTS = Asian koel in long-tailed shrike nest (n = 16 eggs from 4 nests). Spectral reflectance was measured at three egg regions, curves denoted by: b = blunt spectral reflectance of the egg, m = middle spectral reflectance of the egg, t = top spectral reflectance of the egg.

response variables in the analysis to assess variation in koel egg pattern and whether this could be attributed to egg mimicry for the three host species. *Maximum energy*, which corresponds to the dominant marking size on the measured egg, is the energy at the maximum frequency or dominant marking filter size. *Proportion energy*, the proportion of the total energy across all scales, is a measure of the diversity of pattern sizes or how much the main pattern size dominates. A high value shows that the egg pattern is dominated by the main spot size. The proportion energy is calculated by the maximum energy divided by the total energy. *Total energy* is the sum of the pattern energies at every scale or amplitude, measuring overall pattern contrast

against the egg background color. *Dispersion* is measured as the standard deviation of the energy values across all scales. This measure represents the variation of marking sizes across the egg [10].

**Egg color and luminance.** In 2015 we measured egg color and luminance of a sample of Asian koel and host eggs using an Ocean Optics Jaz spectrophotometer, a narrow-ended UV-Vis unidirectional Ocean Optics QR400-7-SR reflectance probe with a 5 mm diameter, and an Ocean Optics WS-1 reflectance standard following the methods of Abernathy et al. [42]. We measured the spectral reflectance of eggs in the range 300–700 nm. Before measuring an egg, a dark and white standard reference was taken. Egg measurements were taken under a black cloth in the field to reduce noise from ambient light. Three measurements were taken from a random location within each of the three egg regions (blunt, middle and top) for each egg. Spectral reflectance of each egg region was attained by averaging all three measurements per region.

Due to the fact that human and avian visual systems differ significantly, avian visual modelling was used to determine the similarity of egg appearance between Asian koel and host eggs chromatically (color) and achromatically (luminance) according to the host's visual perspective [43]. We measured the cone stimulation (photon catch) values for the violet sensitive (VS), ultraviolet sensitive (UVS), shortwave sensitive (SWS), mediumwave sensitive (MWS), longwave sensitive (LWS) and double cones (luminance) using the "pavo" package [80] in the R Statistical Package [81]. Moreover, we performed a just-noticeable-differences (JNDs) analysis, comparing the color and luminance of every Asian koel egg to every host clutch as well as Asian koel eggs from other host nests [following 82]. JND values less than 1 indicate that two eggs are indistinguishable. JND values between 1–3 indicate that the eggs are barely distinguishable in ideal lighting conditions, while the eggs should be easily distinguishable in good lighting conditions when the JNDs are more than three [83].

Birds have two distinct visual systems. Species with an ultraviolet sensitive (UVS) visual system are able to view most of the ultraviolet visual range (300–400 nm), while species with a violet sensitive (VS) visual system are only able to view part of the ultraviolet visual range (340–400 nm) [84]. Among *Corvus* spp. (crows) that have been tested, all have a VS visual system, while all tested Sturnidae (starling, myna, and others) species, including the common myna, have a UVS visual system [85–89]. It is unclear what the visual system of shrikes is, as some studies have found evidence for a UVS visual system [e.g. 90], while another study found evidence for a VS visual system [91]. Therefore, we analyzed all the species using both the UVS visual system of the blue tit (*Cyanistes caeruleus*) and the VS visual system of the common peafowl (*Pavo cristatus*) [92,93].

## Statistical analysis

We examined variation among Asian koel eggs from specific host nests and their host eggs using the following egg characteristics: egg volume, shape, four pattern variables, reflectance, and both achromatic and chromatic cone stimulation values and JNDs. Each egg character was fitted with a linear mixed-effects model, assessing Asian koel and host eggs separately. The egg character was used as the response variable, and egg type (categorical with 3 levels, i.e. three Asian koel egg types or three host egg types) was added as a fixed effect. Year was added with random intercepts, and for host eggs we added random intercepts for nest identity to account for non-independence. Due to some deviations from normal distribution of residuals from the linear mixed-effects models, we ran additional sets of analyses, running identical models using instead log-transformed values. The models using log-transformed values gave qualitatively similar results, thus corroborating our findings (see, S1 Table). Additionally, we

ran linear mixed-effects models to examine differences between Asian koel eggs from specific host nests compared to the eggs of their host species (see, S2 and S3 Tables).

Moreover, we examined color variation in Asian koel eggs laid in long-tailed shrike and common myna nests using cone stimulation values as the response variable in linear mixed-effects models. Unfortunately, no parasitized house crow nests were found during the period when we had access to a spectrophotometer, so we were unable to collect spectral reflectance data of Asian koel eggs from house crow nests. We added egg type (categorical with two levels, Asian koel egg type, i.e. from common myna and long-tailed shrike nests) and egg region (categorical with three levels; blunt, middle, and top) as fixed effects. Random intercepts were added for egg identity to account for non-independence within the same egg. We combined both egg region and egg type together to test for differences among Asian koel eggs accounting for differences among egg regions. Finally, we examined pair-wise differences in average chromatic (color) and achromatic (luminance) JNDs (from comparisons of each Asian koel egg with the respective host clutch and also with every other host clutch) using Dunn post hoc tests. We accounted for multiple comparisons following Benjamini and Hochberg's procedure for false discovery rate (FDR) [94] (S4 Table). All statistical analyses were carried out using the R Statistical Package v. 3.6.3 [81], linear mixed-effects models were fitted using the R-package '*glmmTMB*' [95], and model fit was evaluated using the '*DHARMa*'-package [96].

## Results

### Egg volume and shape

The mean (± SD) length and breadth of the house crow, common myna and long-tailed shrike eggs were 37.33 ± 2.05 × 26.54 ± 1.15 mm, 28.39 ± 0.94 × 20.73 ± 0.98 mm and 22.85 ± 0.89 × 18.10 ± 0.54 mm, respectively. Similarly, the mean (± SD) length and breadth of Asian koel eggs laid in the house crow, common myna and long-tailed shrike nests were 29.96 ± 1.09 × 22.98 ± 0.67 mm, 30.42 ± 1.32 × 22.71 ± 1.18 mm and 31.17 ± 1.26 × 23.43 ± 0.83 mm, respectively. All Asian koel egg types were different from their corresponding host egg type in volume and shape (S2 Table). The egg volume of Asian koel eggs laid in different host nests differed significantly. However, differences did not match the egg volume of the corresponding host species (Table 1; Fig 2). Significantly larger Asian koel eggs were found in long-tailed shrike nests compared to Asian koel eggs laid in nests of the two other host species (Table 1; Fig 2). Asian koel eggs laid in house crow nests were less elongated compared to those laid in common myna nests, but that did not correspond to the egg shape of its host species (Table 1; Fig 2). Moreover, egg volume and shape differed significantly among the three host species (Table 1). The house crow eggs were significantly more elongated than the other two hosts and common myna eggs were significantly more elongated than shrike eggs (Fig 2). Altogether, small differences in volume and shape between Asian koel eggs did not correspond to the differences between their respective hosts, suggesting the absence of mimicry (Fig 2).

### Egg pattern variables

Asian koel and host egg types differed significantly in three out of four egg pattern characteristics, where only in the proportion energy we did not find any difference between Asian koel eggs in house crow nests and house crow eggs (S2 Table). There were no differences in either of the four pattern variables among Asian koel eggs in different host nests (Table 1, Fig 3). Running identical models using log-transformed values, improving normality of residuals from each model, corroborated the analyses using non-transformed values. Analyses on log-transformed values showed larger *maximum energy* for Asian koel eggs in house crow nests than Asian koel eggs in common myna nests (S1 Table; Fig 3). Asian koel eggs in house crow

**Table 1. Linear mixed-effects models on egg characteristics (volume, shape and spotting pattern) of Asian koel eggs and three host species (common myna, house crow and long-tailed shrike).**

| Egg type | n (eggs/nests) | Egg characteristic | Estimate | SE | z | P | |
|---|---|---|---|---|---|---|---|
| AK_CM-AK_HC | 167/58, 22/17 | Volume | -90.29 | 239.00 | -0.38 | 0.706 | |
| AK_CM-AK_LTS | 167/58, 82/36 | Volume | 550.80 | 158.10 | 3.48 | <0.001 | *** |
| AK_HC-AK_LTS | 22/17, 82/36 | Volume | 641.08 | 261.08 | 2.46 | 0.014 | * |
| Common Myna-House Crow[1] | 389/128, 304/99 | Volume | 7910.80 | 150.90 | 52.42 | <0.001 | *** |
| Common Myna-Long-tailed Shrike[1] | 389/128, 213/59 | Volume | -2472.20 | 173.80 | -14.22 | <0.001 | *** |
| House Crow-Long-tailed Shrike[1] | 304/99, 213/59 | Volume | -10383.00 | 185.80 | -55.88 | <0.001 | *** |
| AK_CM-AK_HC | 167/58, 22/17 | Shape | -0.04 | 0.01 | -2.59 | 0.010 | * |
| AK_CM-AK_LTS | 167/58, 82/36 | Shape | -0.01 | 0.01 | -1.25 | 0.212 | |
| AK_HC-AK_LTS | 22/17, 82/36 | Shape | 0.03 | 0.01 | 1.77 | 0.076 | |
| Common Myna-House Crow | 389/128, 304/99 | Shape | 0.04 | 0.01 | 4.84 | <0.001 | *** |
| Common Myna-Long-tailed Shrike | 389/128, 213/59 | Shape | -0.11 | 0.01 | -11.47 | <0.001 | *** |
| House Crow-Long-tailed Shrike | 304/99, 213/59 | Shape | -0.15 | 0.01 | -14.56 | <0.001 | *** |
| AK_CM-AK_HC[1] | 36/16, 12/10 | Max energy | 96.74 | 60.20 | 1.61 | 0.108 | |
| AK_CM-AK_LTS[1] | 36/16, 32/12 | Max energy | -20.35 | 43.87 | -0.46 | 0.643 | |
| AK_HC-AK_LTS[1] | 12/10, 32/12 | Max energy | -117.09 | 61.13 | -1.92 | 0.055 | |
| House Crow-Long-tailed Shrike | 107/35, 57/17 | Max energy | 736.70 | 134.40 | 5.48 | <0.001 | *** |
| AK_CM-AK_HC | 36/16, 12/10 | Prop energy | 0.01 | 0.00 | 1.52 | 0.128 | |
| AK_CM-AK_LTS | 36/16, 32/12 | Prop energy | -0.00 | 0.00 | -0.08 | 0.935 | |
| AK_HC-AK_LTS | 12/10, 32/12 | Prop energy | -0.01 | 0.00 | -1.56 | 0.120 | |
| House Crow-Long-tailed Shrike[1] | 107/35, 57/17 | Prop energy | 0.02 | 0.00 | 4.62 | <0.001 | *** |
| AK_CM-AK_HC[1] | 36/16, 12/10 | Sum energy | 783.30 | 546.10 | 1.43 | 0.152 | |
| AK_CM-AK_LTS[1] | 36/16, 32/12 | Sum energy | -211.50 | 398.10 | -0.53 | 0.595 | |
| AK_HC-AK_LTS[1] | 12/10, 32/12 | Sum energy | -994.80 | 554.60 | -1.79 | 0.073 | |
| House Crow-Long-tailed Shrike | 107/35, 57/17 | Sum energy | 3618.8.80 | 986.20 | 3.67 | <0.001 | *** |
| AK_CM-AK_HC[1] | 36/16, 12/10 | SD energy | 17.99 | 16.55 | 1.09 | 0.277 | |
| AK_CM-AK_LTS[1] | 36/16, 32/12 | SD energy | -9.47 | 12.06 | -0.79 | 0.433 | |
| AK_HC-AK_LTS[1] | 12/10, 32/12 | SD energy | -27.45 | 16.81 | -1.63 | 0.102 | |
| House Crow-Long-tailed Shrike | 107/35, 57/17 | SD energy | 210.74 | 35.91 | 5.87 | <0.001 | *** |

[1]Residuals deviating from normal distribution.

AK_CM = Asian koel in common myna nests, AK_HC = Asian koel in house crow nests and AK_LTS = Asian koel in long-tailed shrike nests, and each host respectively. Asian koel eggs and host eggs were analyzed in separate models. For identical models run on log-transformed values, see S1 Table.

nests showed larger *dispersion* than Asian koel eggs from long-tailed shrike nests (S1 Table; Fig 3). However, none of these differences remained significant when using non-transformed values (Fig 3). In contrast to Asian koel eggs, host eggs of house crow and long-tailed shrike differed significantly in all pattern variables (Fig 3).

## Eggshell reflectance and color

The mean spectral reflectance of Asian koel eggs from common myna and long-tailed shrike nests did not appear to resemble the reflectance of any of the hosts (Fig 1). The JND analysis showed that Asian koel eggs from different host nests were barely distinguishable (JNDs = 1–3) from one another in both chromatic and achromatic JNDs for both a UVS and VS visual system (Figs 4 and 5). However, Asian koel eggs were easily distinguishable (JNDs > 3) from common myna and house crow eggs in good lighting conditions for both visual systems, but were not easily distinguishable (JNDs = 1–3) from long-tailed shrike eggs in all egg regions for

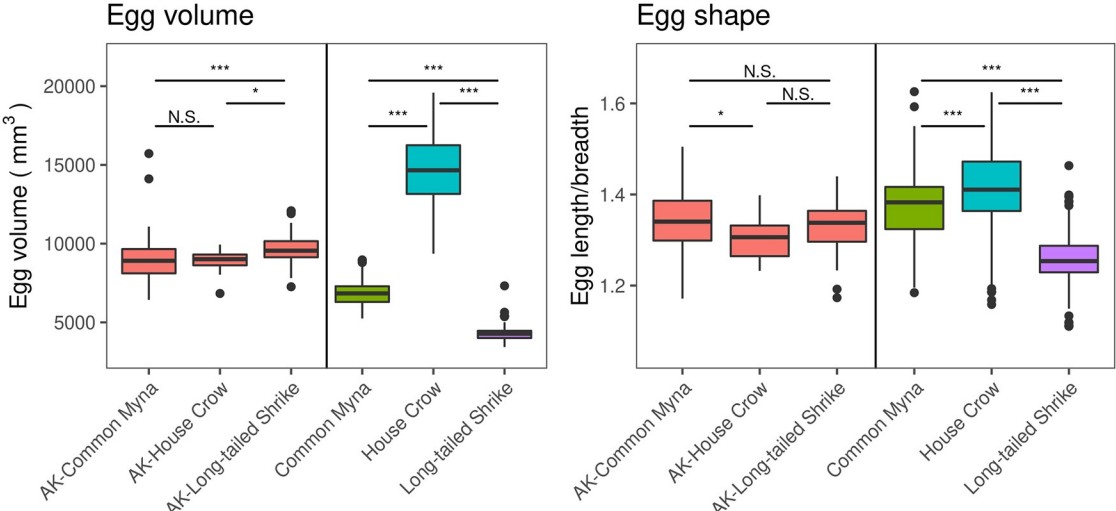

**Fig 2. Egg volume and shape of Asian koel (AK) eggs laid in specific host nests (left panels) and eggs of host species (right panel).** Asterisks denotes significant difference among Asian koel eggs and host eggs (see Table 1; N.S. p > 0.05, *p < 0.05, **p < 0.01, ***p < 0.001). Host and Asian koel eggs were analyzed separately. Vertical bars depict the 95% confidence interval.

chromatic JNDs in the VS system (Figs 4 and 5). Asian koel eggs laid in common myna and long-tailed shrike nests did not differ significantly in any of cone stimulation values for color or luminance cones for either the UVS or VS visual system models (Table 2). All regions of Asian koel eggs were significantly different from each other (except for the ultraviolet cone in the UVS system, where the blunt and middle region were not statistically different; Table 2). Altogether, these results suggest little resemblance between Asian koel and host eggs in color, and we thus find little evidence for mimicry in this host-parasite system.

## Discussion

We investigated different egg parameters, including egg volume, shape, spotting pattern and color (using avian visual modelling) to examine Asian koel host-specific egg mimicry. Our results revealed that there were only a few differences between Asian koel eggs laid in different host nests in egg volume and shape, and no differences in egg spotting pattern variables or color. In contrast, we found large variation among host eggs when examining the same egg characteristics. Comparisons to host egg variation further suggest that the small variation among Asian koel eggs was unlikely to comprise mimicry of host eggs.

Due to the natural selection in host-specific co-evolutionary adaptations (i.e. arms race), the parasite egg morphology may converge to the host egg morphology, changing egg morphology away from the ancestral morphology of the parasite egg. The most closely related species of the Asian koel, the Pacific koel, lays eggs that are pinkish but sparingly spotted and blotched, especially on the blunt end with chestnut and purplish brown [42]. This egg morph is distinctly different from the Asian koel eggs. A previous study revealed that Pacific koels evolved egg mimicry of one of the main hosts, the noisy friarbird (*Philemon corniculatus*), which allowed the Pacific koel to exploit new hosts with similar egg morphologies [42]. Asian koel eggs were previously believed to be mimetic to house crow eggs, but the present study did not find any support for this. Further studies on a regional basis are needed to investigate in more detail if Asian koel egg morphs vary according to host use throughout Asia. Altogether we did not find evidence of egg mimicry in the present study, which may be a result of a lack

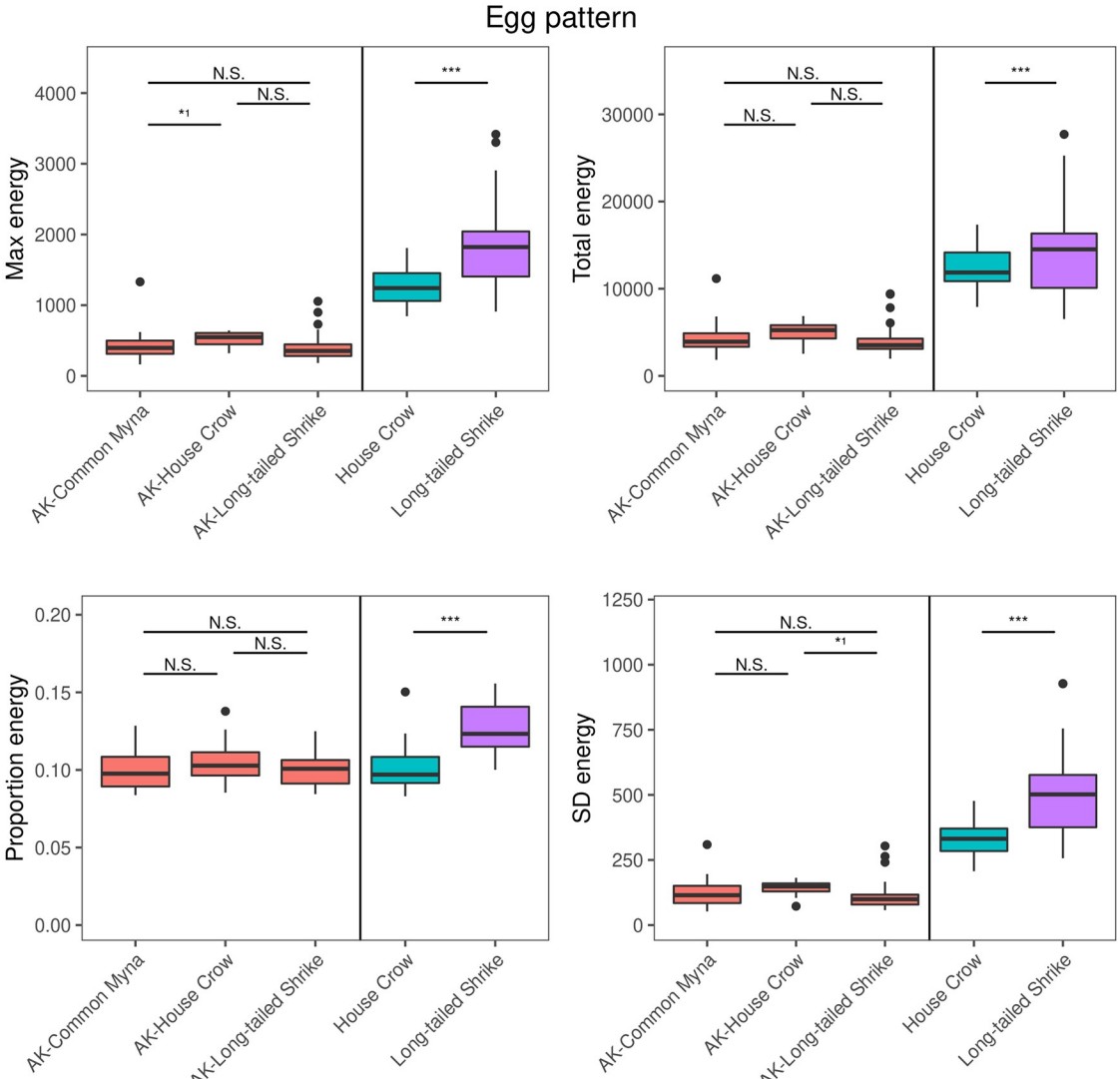

**Fig 3. Egg spotting pattern variables of Asian koel (AK) eggs laid in the nests of three host species (left panel) and two host species (house crow and long-tailed shrike; right panel).** Asterisks denotes significant difference between Asian koels and hosts, respectively, based on linear-mixed effects model outputs (see Table 1; N.S. p > 0.05, *p < 0.05, **p < 0.01, ***p < 0.001). Tests of Asian koel and hosts eggs were performed separately. Vertical bars depict the 95% confidence interval. [1]Significant only after log-transformation (see S1 Table).

of rejection response to Asian koel eggs among the three hosts studied, and/or potential absence of host-specific parasitism. The variation of Asian koel eggs in some egg traits is probably due to individual genetic differences in the Asian koel population or possibly due to the Asian koel mating system. Unfortunately, we do not possess any molecular data, which might help to explain the results. Future genetic analyses or telemetry studies may reveal more information about the source of the variation found between Asian koel eggs.

However, it is possible that the observed variation of Asian koel eggs may be a result of predation rather than parasitism (i.e. eggs have evolved to become cryptic rather than mimetic). Recent studies have found that house crows that lay immaculate blue eggs had higher (75%, n = 4) nest predation (sample size too small to be tested statistically) than regular crow eggs with a bluish-green ground color and black or brown blotches (28.3%, n = 60) [73]. In

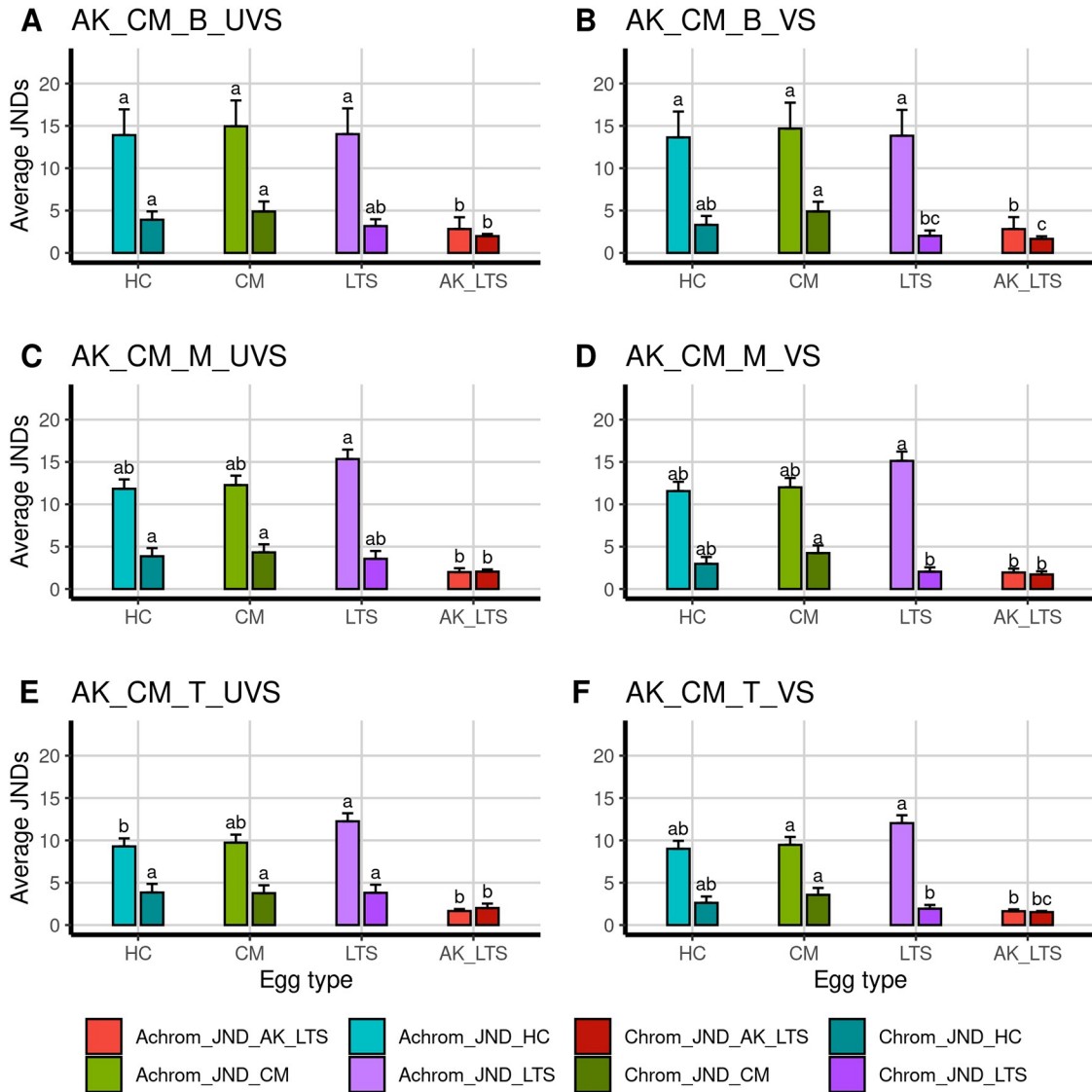

**Fig 4. Average achromatic and chromatic JNDs ± SD between Asian koel eggs laid in common myna nests (AK_CM) compared to three hosts: The house crow (HC), the common myna (CM) and the long-tailed shrike (LTS) and the eggs of the Asian koel laid in long-tailed shrike nests (AK_LTS).** Achromatic (Achrom_JND) and chromatic (Chrom_JND) analyses were performed for egg regions (B = blunt, M = middle, T = top) and visual system (UVS = ultraviolet-sensitive, VS = violet-sensitive) separately. Letters above columns denote significant differences (Dunn post-hoc test, p < 0.05 after correcting for multiple tests following Benjamini and Hochberg 1995 [94], see S4 Table).

addition, the house crow is a poor egg rejecter and Asian koel and house crow eggs only matched in one pattern variable, proportion energy. Therefore, the selective force behind the apparent resemblance of Asian koel and crow eggs could be due to nest predation instead of crow egg rejection, but this needs more investigation, and the support for the influence of predation on the evolution of parasitic eggs is quite scarce [9,97,98].

Imperfect adaptation in terms of host egg discrimination and cuckoo egg mimicry are often a result of time lag in the evolution of traits [evolutionary lag hypothesis, see 2, 6, 99], occurrence of recognition errors and rejection costs [evolutionary equilibrium hypothesis, see 34, 100, 101] or absence of a later line defense when the prior line defense is successful [strategy

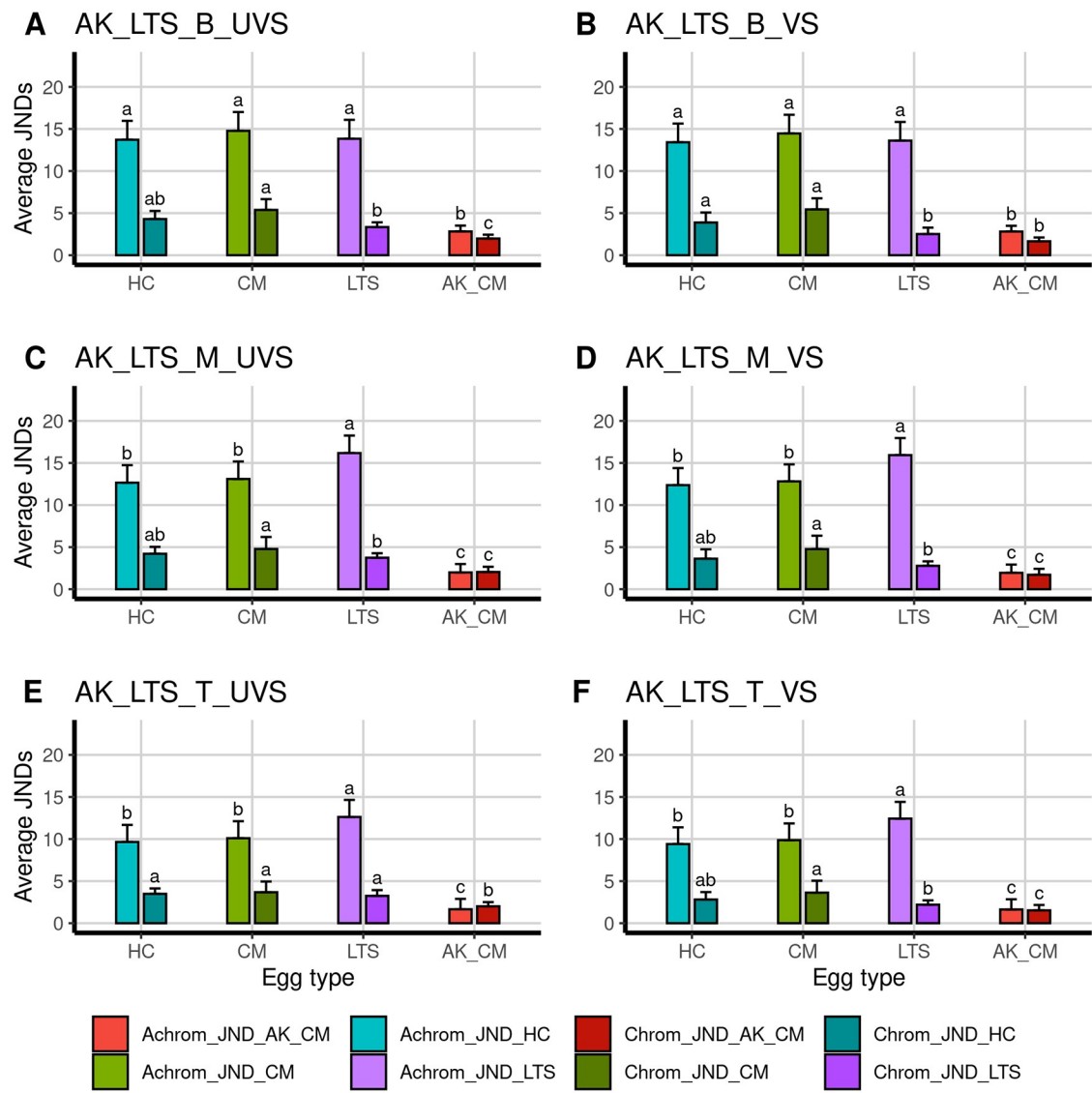

**Fig 5. Average achromatic and chromatic JNDs ± SD between Asian koel eggs laid in long-tailed shrike nests compared to three hosts: The house crow (HC), the common myna (CM) and the long-tailed shrike (LTS) and the eggs of the Asian koel laid in common myna nests (AK_CM).** Achromatic (Achrom_JND) and chromatic (Chrom_JND) analyses were performed for each egg region (B = blunt, M = middle, T = top) and visual system (UVS = ultraviolet-sensitive, VS = violet-sensitive), separately. Letters above columns denote significant differences (Dunn post-hoc test, p < 0.05 after accounting for multiple tests following Benjamini and Hochberg 1995 [94], see supplement for tests S4 Table).

blocking hypothesis, see 102, 103]. The house crow was cited as a host in 370 A.D [1,59], but still displays poor egg discrimination behavior, providing low support for the evolutionary lag hypothesis [69]. Shrikes might accept foreign eggs due to rejection costs, if there is a risk of breaking their own eggs while trying to reject the larger Asian koel eggs. We sometimes observed damaged host eggs in parasitized clutches, however, it is also possible that the host egg damage was caused by the Asian koel during egg laying [laying damage hypothesis, 104, 105]. Furthermore, there is currently no information on how hosts would respond to adult Asian koel encounters by the nest or whether Asian koel chicks are discriminated in the nest. Although, no mobbing behavior towards adult Asian koel was observed, nor any

**Table 2. Linear mixed effects model using cone stimulation values of Asian koel eggs as response, and host species (common myna and long-tailed shrike) and in different egg regions (blunt, middle, top) as fixed effects.**

| Visual system | Egg type and region | Cone stimulation type | Estimate | SE | Df | t | P |
|---|---|---|---|---|---|---|---|
| UVS | AK_CM-AK_LTS | U | 0.002 | 0.003 | 19 | 0.90 | 0.367 |
| | Blunt-Middle | U | 0.002 | 0.002 | 40 | 0.98 | 0.329 |
| | Blunt-Top | U | 0.014 | 0.002 | 40 | 8.76 | <0.001 |
| | Middle-Top | U | 0.012 | 0.002 | 40 | 7.78 | <0.001 |
| | AK_CM-AK_LTS | S | -0.004 | 0.011 | 19 | -0.39 | 0.694 |
| | Blunt-Middle | S | 0.015 | 0.005 | 40 | 2.93 | 0.003 |
| | Blunt-Top | S | 0.046 | 0.005 | 40 | 9.01 | <0.001 |
| | Middle-Top | S | 0.031 | 0.005 | 40 | 6.07 | <0.001 |
| | AK_CM-AK_LTS | M | -0.004 | 0.011 | 19 | -0.37 | 0.712 |
| | Blunt-Middle | M | 0.019 | 0.006 | 40 | 3.24 | 0.001 |
| | Blunt-Top | M | 0.056 | 0.006 | 40 | 9.47 | <0.001 |
| | Middle-Top | M | 0.037 | 0.006 | 40 | 6.23 | <0.001 |
| | AK_CM-AK_LTS | L | -0.002 | 0.006 | 19 | -0.33 | 0.741 |
| | Blunt-Middle | L | 0.010 | 0.004 | 40 | 2.43 | 0.015 |
| | Blunt-Top | L | 0.034 | 0.004 | 40 | 8.68 | <0.001 |
| | Middle-Top | L | 0.025 | 0.004 | 40 | 6.26 | <0.001 |
| | AK_CM-AK_LTS | Luminance | -0.003 | 0.008 | 19 | -0.37 | 0.709 |
| | Blunt-Middle | Luminance | 0.015 | 0.005 | 40 | 3.05 | 0.002 |
| | Blunt-Top | Luminance | 0.047 | 0.005 | 40 | 9.38 | <0.001 |
| | Middle-Top | Luminance | 0.032 | 0.005 | 40 | 6.33 | <0.001 |
| VS | AK_CM-AK_LTS | V | -0.002 | 0.006 | 19 | -0.41 | 0.680 |
| | Blunt-Middle | V | 0.007 | 0.003 | 40 | 2.53 | 0.012 |
| | Blunt-Top | V | 0.028 | 0.003 | 40 | 9.32 | <0.001 |
| | Middle-Top | V | 0.020 | 0.003 | 40 | 6.80 | <0.001 |
| | AK_CM-AK_LTS | S | -0.004 | 0.013 | 19 | -0.33 | 0.743 |
| | Blunt-Middle | S | 0.019 | 0.006 | 40 | 3.09 | 0.002 |
| | Blunt-Top | S | 0.055 | 0.006 | 40 | 9.02 | <0.001 |
| | Middle-Top | S | 0.036 | 0.006 | 40 | 5.93 | <0.001 |
| | AK_CM-AK_LTS | M | -0.004 | 0.011 | 19 | -0.38 | 0.708 |
| | Blunt-Middle | M | 0.019 | 0.006 | 40 | 3.24 | 0.001 |
| | Blunt-Top | M | 0.055 | 0.006 | 40 | 9.47 | <0.001 |
| | Middle-Top | M | 0.036 | 0.006 | 40 | 6.24 | <0.001 |
| | AK_CM-AK_LTS | L | -0.002 | 0.006 | 19 | -0.34 | 0.735 |
| | Blunt-Middle | L | 0.010 | 0.004 | 40 | 2.45 | 0.014 |
| | Blunt-Top | L | 0.035 | 0.004 | 40 | 8.71 | <0.001 |
| | Middle-Top | L | 0.025 | 0.004 | 40 | 6.26 | <0.001 |
| | AK_CM-AK_LTS | Luminance | -0.003 | 0.009 | 19 | -0.39 | 0.700 |
| | Blunt-Middle | Luminance | 0.016 | 0.005 | 40 | 3.07 | 0.002 |
| | Blunt-Top | Luminance | 0.048 | 0.005 | 40 | 9.37 | <0.001 |
| | Middle-Top | Luminance | 0.033 | 0.005 | 40 | 6.30 | <0.001 |

AK_CM = Asian koel eggs in common myna nests, AK_LTS = Asian koel eggs in long-tailed shrike nests, U = ultraviolet-sensitive, V = violet-sensitive, S = shortwave-sensitive, M = mediumwave-sensitive, L = longwave-sensitive. UVS/VS = Ultraviolet or violet sensitive visual system.

discrimination against Asian koel chicks recorded during the study, thorough experiments would be needed to properly test the strategy blocking hypothesis [102,103].

Another possible explanation for the lack of mimicry of host eggs by Asian koels is that host tolerance of Asian koel eggs could relax selection for egg mimicry [103,106–108], however, this implies stronger egg rejection in the past or in other hosts. It is possible that hosts of the Asian koel can minimize the cost of parasitism by adjusting some life history or reproductive traits, such as increasing or decreasing clutch size, raising multiple broods in a breeding season, providing more maternal investment to their own eggs, or accelerating their own nestling development [106,107]. Tolerance of parasitic eggs can be adaptive when a brood parasite is less virulent, like the non-evicting Asian koel [103]. However, there is no information on whether host tolerance exists in hosts of the Asian koel, so future studies must be conducted on this type of host defense.

The Asian koel eggs seemed to match common myna eggs slightly better than the two other hosts in both egg volume and shape, although they are still significantly different from their host in these parameters (see, S2 Table). Additionally, Asian koel eggs are spotted, while common myna eggs are immaculate. It is likely that the closer matching of Asian koel eggs with common myna eggs in volume and shape has occurred by chance, as there is little evidence of selection for mimicry by mynas; they routinely accept Asian koel eggs and show poor discrimination of blue and brown model eggs [69]. However, we cannot exclude the possibility that Asian koels have evolved cryptic eggs in common myna nests. Some brood parasites lay cryptic eggs instead of mimetic eggs in dark, domed nests, which makes it difficult for the host to recognize the parasite eggs in the nest [52–54]. As mentioned previously, common myna nests often have poor lighting conditions due to nest structure and position (domed nests or placed inside holes or cavities). It is therefore possible that the closer matching of common myna and host-specific Asian koel eggs in egg volume, shape and grey-bluish egg color with numerous brown and black spots of Asian koel eggs in dim light conditions may be an adaptation of the Asian koel that increases acceptance of their eggs by the host. Recent visual modeling techniques revealed that immaculate, matt dark olive or brown eggs are difficult to detect by hosts in darker domed nests [53,109]. Hosts with dark nest interiors typically discriminate against foreign eggs using egg size and shape as cues [18,19,22,110,111, but see 112,113]. Moreover, cryptic eggs can be an advantage when multiple parasitism by different parasitic females is common, like at our study site, if female parasites remove host or other parasitic eggs while depositing their own egg [27,52,54]. While there is no information about whether Asian koel females selectively remove other cuckoo rather than host eggs from a nest during egg laying, this behavior has been shown to occur in greater honeyguides (*Indicator indicator*) [114] and little bronze-cuckoos (*Chrysococcyx minutillus*) [52], but not in common cuckoos [115].

Spotting pattern of Asian koel eggs laid in different host nests show general similarities in maximum energy, proportion energy, total energy and dispersion. Among the host species, the common myna lays immaculate blue eggs, which appear different from Asian koel eggs in pattern to the human eye. Although the general assumption is that Asian koels are mimicking crow eggs [57,58,66], we found no support for this assumption. This finding is not surprising, given that poor rejection rates of immaculate blue and brown model eggs by the house crow suggest that the host may not discriminate against foreign eggs well based on color or spotting and, thus, there is probably either no or limited selection pressure on the Asian koel to mimic crow eggs [see 69].

In our avian visual models for the egg color analysis, we found no significant differences between Asian koel eggs laid in common myna and long-tailed shrike nests and we did not have any egg color data of Asian koel eggs laid in house crow nests. However, we did find that the color of the egg regions of Asian koel eggs differed significantly. The JND and cone

simulation analyses of Asian koel and host eggs suggest that the Asian koel has not evolved host-specific egg colors.

Among the three host species, the house crow laid the largest eggs, while the long-tailed shrike laid the smallest eggs, but the largest Asian koel eggs were found in long-tailed shrike nests. Host egg discrimination behavior drives selection on cuckoos to evolve host egg mimicry [6,9,30,79], thus, any ongoing selection for the evolution of egg volume should not have resulted in the largest Asian koel eggs being laid in long-tailed shrike nests. Using immaculate blue and brown model eggs, Begum et al. [69] showed that house crows reject only 9.1% (n = 2 out of 22) of model eggs, common mynas reject 0% of model eggs (n = 0 out of 22), and long-tailed shrikes reject 75% (n = 15 out of 20) of model eggs. Thus, shrikes appear to be capable of rejecting odd eggs, but they regularly accept Asian koel eggs [69]. One explanation may be that it is difficult for shrikes to grasp and eject the larger Asian koel eggs (31.17 mm ± 1.26 SD) compared to the model eggs used by Begum et al. [69] (24.96 mm ± 19.29 SD). Alternatively, long-tailed shrikes may be able to puncture and eject the Asian koel eggs when it is difficult to grasp and eject. However, we did not find any punctured or broken Asian koel eggs in and around long-tailed shrike nests. Future study with video recordings may confirm shrike responses to Asian koel eggs. Moreover, in theory, if shrikes can detect Asian koel eggs, they might benefit by abandoning the parasitized clutch and starting a new brood, as parasitism reduces shrike reproductive success [57]. However, since Asian koels are non-evictors and nest predation is high at our study site (50.8% of shrike nests were predated, [71]), the benefits of accepting the Asian koel egg might outweigh the costs, since they are sometimes able to raise their own young with an Asian koel nestling. It may therefore be better for the host to do the "best of a bad job" and accept reduced reproductive output (i.e. lower number of chicks produced in parasitized than non-parasitized nests), rather than risking all reproductive output in a second brooding attempt in which there is a high probability that the nest will be predated.

## Conclusion

We found few differences between Asian koel eggs laid in different host species' nests in egg volume, shape, pattern variables and color. Importantly, the consistent differences among Asian koel eggs in size and shape did not match the corresponding host species, thus we found no evidence of host-specific egg mimicry in the three host species studied. Potential causes might include that these host species show poor rejection behavior, or that the Asian koel may not be host specific. The underlying mechanisms for the lack of egg mimicry need further investigation.

## Supporting information

**S1 Table. Linear mixed effects model using log-transformed egg characteristics as response (log(volume), log(shape) and log pattern variables), for Asian koel eggs and three host species (common myna, house crow and long-tailed shrike).** Asian koel eggs and host eggs were assessed separately, and host egg were accounted for non-independence. AK_CM = Asian koel in common myna nests, AK_HC = Asian koel in house crow nests and AK_LTS = Asian koel in long-tailed shrike nests.
(DOCX)

**S2 Table. Model outputs from linear mixed-effects models showing differences between Asian koel eggs from specific host nests and their corresponding host eggs (see Figs 2 and 3).** Each egg characteristic was used as response, and koel vs. host as a fixed factor.
(DOCX)

**S3 Table. Model outputs from linear mixed-effects models showing differences between Asian koel eggs from specific host nests and their corresponding host eggs (see Figs 2 and 3).** Log-transformed values of egg characteristic was used as response, and koel vs. host as a fixed factor, to improve normality of residuals (see, S2 Table).
(DOCX)

**S4 Table. Model outputs from Dunn post host tests, showing pair-wise comparisons of differences in JNDs between Asian koel eggs in common myna nests (egg = AK_CM) and long-tailed shrike nests (egg = AK_LTS).** P-values were adjusted (P.adj.) following Benjamini and Hochberg. Significant differences are denoted with letters in Figs 4 and 5.
(DOCX)

**S1 Data. Data file for egg volume and shape.** Egg morphology data comprising data from Asian koel, and three host species. Length and breadth in mm. Location: Jahangirnagar University campus, Bangladesh. Year: 2008–2013 and 2015–2017.
(CSV)

**S2 Data. Data file for egg spotting pattern.** Egg pattern data comprising data from Asian koel, and three host species. Location: Jahangirnagar University campus, Bangladesh. Data collection year: 2015–2017.
(CSV)

**S3 Data. Data file for egg spectral reflectance.** Location: Jahangirnagar University campus, Bangladesh. Data collection year: 2015.
(XLSX)

**S4 Data. Data file for Just Noticeable Differences (JNDs).** Asian koel eggs from common myna nests, compared to three host species, and Asian koel eggs from long-tailed shrike nests. Location: Jahangirnagar University campus, Bangladesh. Data collection year: 2015.
(CSV)

**S5 Data. Data file for Just Noticeable Differences (JNDs).** Asian koel eggs from long-tailed shrike nests, compared to three host species, and Asian koel eggs from common myna nests. Location: Jahangirnagar University campus, Bangladesh. Data collection year: 2015.
(CSV)

## Acknowledgments

We thank Mr. Monoronjon Baroi and Mr. Yousuf for their great help with the data collection in the field. We also thank the editor Petr Heneberg and John M. Eadie and three anonymous reviewers for constructive comments that significantly improved the manuscript. We acknowledge Jolyon Troscianko for his advice on the egg pattern analysis.

## Author Contributions

**Conceptualization:** Mominul Islam Nahid, Frode Fossøy, Bård G. Stokke, Virginia Abernathy, Sajeda Begum, Naomi E. Langmore, Eivin Røskaft, Peter S. Ranke.

**Data curation:** Mominul Islam Nahid, Frode Fossøy, Bård G. Stokke, Virginia Abernathy, Eivin Røskaft, Peter S. Ranke.

**Formal analysis:** Mominul Islam Nahid, Frode Fossøy, Bård G. Stokke, Virginia Abernathy, Eivin Røskaft, Peter S. Ranke.

**Funding acquisition:** Eivin Røskaft.

**Investigation:** Mominul Islam Nahid, Bård G. Stokke, Virginia Abernathy, Sajeda Begum.

**Methodology:** Mominul Islam Nahid, Frode Fossøy, Bård G. Stokke, Virginia Abernathy, Naomi E. Langmore, Eivin Røskaft, Peter S. Ranke.

**Supervision:** Frode Fossøy, Bård G. Stokke, Eivin Røskaft.

**Visualization:** Mominul Islam Nahid, Peter S. Ranke.

**Writing – original draft:** Mominul Islam Nahid.

**Writing – review & editing:** Mominul Islam Nahid, Frode Fossøy, Bård G. Stokke, Virginia Abernathy, Sajeda Begum, Naomi E. Langmore, Eivin Røskaft, Peter S. Ranke.

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
