## [Decision Letter · Decision Letter 0]

27 Apr 2021

PONE-D-21-04869

No evidence of host-specific egg mimicry in Asian koels Eudynamys scolopaceus

PLOS ONE

Dear Dr. Nahid,

Thank you for submitting your manuscript to PLOS ONE. After careful consideration, we feel that it has merit but does not fully meet PLOS ONE’s publication criteria as it currently stands. Therefore, we invite you to submit a revised version of the manuscript that addresses the points raised during the review process.

We look forward to receiving your revised manuscript.

Kind regards,

Petr Heneberg

Academic Editor

PLOS ONE

Journal Requirements:

2. Thank you for stating the following in the Acknowledgments/Funding Section of your manuscript:

The study was funded by grants through ‘Quota Scheme’ at Norwegian University of Science

and Technology (NTNU), from NTNU, Department of Biology to Professor Eivin Røskaft and

an Australian National University travel grant to Virginia Abernathy. The funders had no role in

study design, data collection and analysis, decision to publish, or preparation of the manuscript.

The funders had no role in study design, data collection and analysis, decision to publish, or preparation of the manuscript

Reviewers' comments:

Reviewer's Responses to Questions

**Comments to the Author**

1. Is the manuscript technically sound, and do the data support the conclusions?

Reviewer #1: Yes

Reviewer #2: Yes

Reviewer #3: Yes

Reviewer #4: Yes

2. Has the statistical analysis been performed appropriately and rigorously? 

Reviewer #1: No

Reviewer #2: Yes

Reviewer #3: Yes

Reviewer #4: Yes

3. Have the authors made all data underlying the findings in their manuscript fully available?

Reviewer #1: Yes

Reviewer #2: Yes

Reviewer #3: Yes

Reviewer #4: Yes

4. Is the manuscript presented in an intelligible fashion and written in standard English?

Reviewer #1: Yes

Reviewer #2: Yes

Reviewer #3: Yes

Reviewer #4: Yes

5. Review Comments to the Author

Reviewer #1: Comments on PONE-D-21-04869

Title: No evidence of host-specific egg mimicry in Asian koels Eudynamys scolopaceus

In my opinion, this manuscript was well written and made a good contribution to research area of avian brood parasitism.

Thus I have only very minor comments.

1. title:

I suggest “No evidence of host-specific egg mimicry in Asian koels” as title, as most titles do not necessarily have a Latin name.

2. L80:

delete [Nahid et al submitted,], as [56] is ok.

The same as L125-126.

3. L81:

Delete “and may be the first brood parasite mentioned in ancient literature, as it” and changed to “which was documented…”

4. L93:

Pacific koel; and check throughout your text for this.

5. L94-95:

Please use “Asian koels” throughout your text.

6. L275, 330:

Table 1 and 2 should be in three-line tables.

7. L385-386:

In addition to ref. 104-106, here, Yang et al. (2020) should be cited.

8. L450-453:

Please use “see”, not “see-”

9. L478:

What about “Md.”? Perhaps Md. Yousuf should follow “Monoronjon Baroi”

Reviewer #2: In this study, the authors test whether the Asian koel has evolved host-specific egg mimicry in three common host species. They combined several methods to evaluate mimicry, including measurements of egg size and shape, egg spotting patterns, and egg color and luminance using spectrophotometry and modeling on both UVS and VS avian visual systems. They did find significant differences in Asian koel eggs laid in different host nests, however, these differences were the opposite of what would be expected from host mimicry. Although this paper would be strengthened by genetic data from koel eggs to determine maternity and individual identity, this paper is well written, the methodology is clear, and it expands the knowledge on koel host use. I have a few comments that the authors might consider incorporating into their manuscript:

1. I would expand the paragraph in the introduction about cryptic eggs (line 72) to introduce the reader more on the idea of why it would be advantageous for parasitic eggs to be harder for other parasites to see, and explain competition between female parasites more in-depth.

2. You mention in the general methodology (line 154-155) that measurements of host eggs in volume, shape, spotting pattern, and color from the same nest were either averaged or that nest identity was added as a random intercept. You only mention using the random intercepts later in the paper in the statistical methods. I would recommend keeping it consistent and only using random intercepts, so as not to obscure variation in these characteristics that might occur within the same nest.

3. In the discussion, you suggest that larger koel eggs might be harder for shrikes to reject, which may explain why koels lay larger eggs into the nests of shrikes. Is it possible that this pattern could also be driven by competition between koels? As stated in the paper, they have the highest parasitism rates out of the three hosts and also have high multiple parasitism rates. Perhaps larger eggs are an adaptation for koel nestlings to outcompete other koel nestlings? It might be useful to mention survival of parasitic nestlings in multiply parasitized nests.

Reviewer #3: Overall, this is a straightforward manuscript examining for evidence of host-specific egg mimicry by the brood parasitic Asian koel. The authors undertake what is a rather thorough and perhaps even exhaustive approach to explore any convergence in size, shape, reflectance/luminance or egg patterns. I am convinced that there is little evidence of egg mimicry. The authors have explored multiple possible dimensions of mimicry and to me, there are no compelling patterns.

I do confess that I was a bit perplexed as to why such an exhaustive and extensive analysis was deemed necessary. I don’t mean to be overly critical and I appreciate the carefulness but I’m wondering if there is some ongoing debate or question that required such a thorough evaluation. A number of previous references were listed indicating that Asian koel do exhibit egg mimicry of their host species but several of those references seem to be somewhat general or anecdotal reports so perhaps this is really just the focused quantitative analysis? Is there a reason why this might be a contentious issue? If there is a more compelling rationale, that needs to be developed in the introduction. If not, then perhaps the manuscript might be shortened to get straight to the point – no evidence of mimicry. I was impressed that the author list on this paper included individuals who were cited previously as having suggested egg-mimicry by Asian koels and I applaud their willingness to re-examine some of their earlier conclusions or assumptions.

There are a few small concerns to mention, but none that are not particularly critical. For example, the authors reported that they were unable to obtain spectral reflectance data of Asian koel eggs laid in house crow nests yet this would seem to be an important comparison given that the house crow is one species that the Asian koel was previously reported to mimic (and with a long history of having parasitized). The authors simply state that due to logistic reasons (not described) they were unable to analyze those eggs but I think this is a bit of a missing gap in the analysis. Again, given the lack of any evidence of mimicry in any of the other comparisons (Asian koel to the house crow eggs collected elsewhere, or comparisons of Asian koel eggs to eggs of other hosts or koel eggs in other host nests) I’m not sure that this missing piece is critical. Nice but perhaps not absolutely necessary.

More generally, it does seem that one might (snarkily) suggest that perhaps ‘much ado is being made about nothing’ given that house crows and common mynas are not ejector species and Asian koel do not evict host eggs, further lowering the cost to the host. Accordingly, why would we expect mimicry to have evolved – there would have been little selection for host-specific mimicry to evolve in these species? That long-tailed strikes do seem to reject model (non-cryptic eggs) and are – at least currently – a frequently-used host is more interesting but again little evidence of mimicry was found. The large size of the Asian koel eggs laid in nests of long-tailed strikes (which have the smallest eggs of the 3 host species) may well be an adaptation to reduce eggs rejection by the hosts, as the authors note, although this is still speculation.

The authors raise the possibility (lines 58 to 62) that perhaps “moderate mimicry” might be maintained in generalist parasites such that parasite eggs do not accurately match those of any single host but do so ‘moderately” allowing the parasite at least some opportunities to parasitize a broader variety of hosts. The question for me , then, is – what constitutes ‘moderate’ mimicry? I don’t have an operational sense of that and it seems to be a rather open ended and vague concept. How would one reject that hypothesis?

My only other comment is simply that the discussion seems rather long (almost 6 pages) given that there was no evidence of mimicry. I realize the authors are being thorough in discussing a myriad of possible reasons why mimicry has not evolved (e.g., including speculations about the mafia hypothesis, imperfect adaptation, predation rather than parasitism being a driving selection force, or that parasitic Asian koels may remove other koel eggs, etc.). These are all interesting and valid, but quite speculative; perhaps too much discussion is devoted to possible reasons why egg-mimicry does not exist, but for which there is little evidence. The discussion could be shortened by a couple of pages without expensive speculation.

In sum, I did not find any major problems with this analysis. The statistics seem to be adequate although I would suggest that perhaps not every koel egg be treated as an independent sample but rather be “nested” within each nest. I realize nest identity was included as a random intercept but I was not sure that would account for non-independence of each koel egg. Again, not a critical issue.

Overall, I think this manuscript will be a useful contribution, not only to clarify that this species does not exhibit host-specific egg mimicry – as apparently has been claimed – but perhaps even more so as an example of a thorough effort to explore many dimensions of egg mimicry for an obligate brood parasite and several of its host species.

Reviewer #4: This study explores variations in brood parasitic (Asian koel) eggs laid in three different host species. The aim of the study is to determine whether the Asian koel, to some extent, mimics the eggs of its host species. Based on analyses of egg volume, shape, spotting patterns and colour (in the context of avian color vision), the authors found little evidence on the existence of egg mimicry in Asian koel eggs. Furthermore, some characteristics of koel eggs (e.g., egg volume) appear to mismatch the phenotype of host eggs.

Overall, the manuscript is well written and provides quite a lot of details about the study system. The Methods section is clear and provides a complete overview on the general procedures and analyses. The Introduction section is a bit long and I feel that authors go overboard with speculative arguments to explain the lack of egg mimicry in Asian koel eggs. The evolution of mimicry in host-parasite systems is closely related to the costs imposed by parasitism and host rejection abilities. In my opinion, authors should focus their discussion on this point and reduce (not completely eliminate) speculative explanations (eg., tolerance and critical eggs), unless they have evidence that these mechanisms work in their systems. Some of my concerns have been addressed by the authors in their responses to reviewers' comments made to the previous version.

Other comments:

Line 48: “parasite mimicry”. Please change to “egg mimicry”.

Line 48-51: Although factors such as "climatic variables" may play a relatively important role in certain systems, it should be clearly emphasized that the evolution of egg mimicry is closely related to the evolution of host defenses (i.e., egg rejection).

Line 78: “koel chick tolerates…”. Do the authors have evidence of tolerance mechanisms in the Asian koel?

Line 70-71: While examination by spectrophotometry often reveals important information on egg mimicry, I wouldn't say that great-spotted cuckoo eggs are "quite different” from magpie eggs (in many cases they may be almost indistinguishable to the human eye). Please soften this statement.

Lines 72-76: Any mention to cryptic eggs should be integrated in the paragraph about egg mimicry. I think such a paragraph is a bit out of place in the line of argument of introduction.

Line 387: Indeed, there is experimental evidence that eggs volume is an important factor determining egg rejection (Soler at al., Relationships between egg-recognition and egg-ejection in a grasp-ejector species. 2017. Plos One 12(2)).

6. PLOS authors have the option to publish the peer review history of their article (what does this mean?). If published, this will include your full peer review and any attached files.

Reviewer #1: No

Reviewer #2: No

Reviewer #3: **Yes: **John M. Eadie

Reviewer #4: No

---

## [Author Response · Author response to Decision Letter 0]

11 Jun 2021

Reviewer #1: Comments on PONE-D-21-04869

Title: No evidence of host-specific egg mimicry in Asian koels Eudynamys scolopaceus

In my opinion, this manuscript was well written and made a good contribution to research area of avian brood parasitism.

Thus I have only very minor comments.

1. title:

I suggest “No evidence of host-specific egg mimicry in Asian koels” as title, as most titles do not necessarily have a Latin name.

Response: We have removed the Latin name from the title. 

2. L80:

delete [Nahid et al submitted,], as [56] is ok.

The same as L125-126.

Response: We have revised this citation. See line 85. However, for lines 125-126 (currently, 129-132), we reported the parasitism rate of the host species during the period 2008-2017, which is not the same rate mentioned in ref no. 56 (Currently ref no. 57).

3. L81:

Delete “and may be the first brood parasite mentioned in ancient literature, as it” and changed to “which was documented…”

Response: We have revised the wording. See lines 86-87.

4. L93:

Pacific koel; and check throughout your text for this.

Response: We have revised the species name on line 98, and throughout the manuscript.

5. L94-95:

Please use “Asian koels” throughout your text.

Response: We have revised the species name, see line 100 and throughout the text.

6. L275, 330:

Table 1 and 2 should be in three-line tables.

Response: We have revised both tables accordingly. See Table 1 and 2 (lines 294-353).

7. L385-386:

In addition to ref. 104-106, here, Yang et al. (2020) should be cited.

Response: As suggested by the reviewer, to revise and reduce some of the speculative arguments from the discussion, we have removed this paragraph.

8. L450-453:

Please use “see”, not “see-”

Response: We have revised this accordingly. See lines 390-394.

9. L478:

What about “Md.”? Perhaps Md. Yousuf should follow “Monoronjon Baroi”

Response: We have revised the names in Acknowledgements. See line 485. 

Reviewer #2: In this study, the authors test whether the Asian koel has evolved host-specific egg mimicry in three common host species. They combined several methods to evaluate mimicry, including measurements of egg size and shape, egg spotting patterns, and egg color and luminance using spectrophotometry and modeling on both UVS and VS avian visual systems. They did find significant differences in Asian koel eggs laid in different host nests, however, these differences were the opposite of what would be expected from host mimicry. Although this paper would be strengthened by genetic data from koel eggs to determine maternity and individual identity, this paper is well written, the methodology is clear, and it expands the knowledge on koel host use. I have a few comments that the authors might consider incorporating into their manuscript:

1. I would expand the paragraph in the introduction about cryptic eggs (line 72) to introduce the reader more on the idea of why it would be advantageous for parasitic eggs to be harder for other parasites to see, and explain competition between female parasites more in-depth.

Response: We have revised this part, see lines 72-82.

2. You mention in the general methodology (line 154-155) that measurements of host eggs in volume, shape, spotting pattern, and color from the same nest were either averaged or that nest identity was added as a random intercept. You only mention using the random intercepts later in the paper in the statistical methods. I would recommend keeping it consistent and only using random intercepts, so as not to obscure variation in these characteristics that might occur within the same nest.

Response: Thank you for pointing this out. In former analyses this was averaged for the (host) egg spotting variables, but in the revised version, multiple eggs within host clutch were included, thus, allowing for using the clutch identity as random intercepts (as we did for volume and shape). This was unfortunately not edited but is now revised. See lines 162-164

3. In the discussion, you suggest that larger koel eggs might be harder for shrikes to reject, which may explain why koels lay larger eggs into the nests of shrikes. Is it possible that this pattern could also be driven by competition between koels? As stated in the paper, they have the highest parasitism rates out of the three hosts and also have high multiple parasitism rates. Perhaps larger eggs are an adaptation for koel nestlings to outcompete other koel nestlings? It might be useful to mention survival of parasitic nestlings in multiply parasitized nests.

Response: We agree that this aspect would be very interesting to examine. Unfortunately, we do not have data on survival probability in relation to koel egg size. With the current data we are unable to track which of the koel eggs that hatched successfully.

Reviewer #3: Overall, this is a straightforward manuscript examining for evidence of host-specific egg mimicry by the brood parasitic Asian koel. The authors undertake what is a rather thorough and perhaps even exhaustive approach to explore any convergence in size, shape, reflectance/luminance or egg patterns. I am convinced that there is little evidence of egg mimicry. The authors have explored multiple possible dimensions of mimicry and to me, there are no compelling patterns.

I do confess that I was a bit perplexed as to why such an exhaustive and extensive analysis was deemed necessary. I don’t mean to be overly critical and I appreciate the carefulness but I’m wondering if there is some ongoing debate or question that required such a thorough evaluation. A number of previous references were listed indicating that Asian koel do exhibit egg mimicry of their host species but several of those references seem to be somewhat general or anecdotal reports so perhaps this is really just the focused quantitative analysis? Is there a reason why this might be a contentious issue? If there is a more compelling rationale, that needs to be developed in the introduction. If not, then perhaps the manuscript might be shortened to get straight to the point – no evidence of mimicry. I was impressed that the author list on this paper included individuals who were cited previously as having suggested egg-mimicry by Asian koels and I applaud their willingness to re-examine some of their earlier conclusions or assumptions.

Response: Thank you for your suggestions. The extensive analysis in our study was motivated by both the theoretical expectation that host egg rejection will select for egg mimicry in brood parasites, and by conflicting evidence about how generalist parasites reconcile selection to mimic the eggs of multiple hosts with different egg phenotypes. We aimed to test these theories in the Asian koel using robust methods that take into account the differences between avian and human vision. This has now been clarified in the abstract and introduction. 

There are a few small concerns to mention, but none that are not particularly critical. For example, the authors reported that they were unable to obtain spectral reflectance data of Asian koel eggs laid in house crow nests yet this would seem to be an important comparison given that the house crow is one species that the Asian koel was previously reported to mimic (and with a long history of having parasitized). The authors simply state that due to logistic reasons (not described) they were unable to analyze those eggs but I think this is a bit of a missing gap in the analysis. Again, given the lack of any evidence of mimicry in any of the other comparisons (Asian koel to the house crow eggs collected elsewhere, or comparisons of Asian koel eggs to eggs of other hosts or koel eggs in other host nests) I’m not sure that this missing piece is critical. Nice but perhaps not absolutely necessary.

Response: We acknowledge the concern. Unfortunately, we only had access to a spectrophotometer for a limited period of time and no parasitized house crow nests were found during that time. We have added this explanation in lines 250-252. While we were unable to collect reflectance data, we were able to compare the similarity of koel eggs to house crow eggs in terms of spotting pattern, volume and shape, and we found no evidence of mimicry. Moreover, for the human eye there was no evidence of mimicry of the color of house crow eggs by koel eggs laid in house crow nests, indicating a lack of mimicry in the human-visible part of the visual spectrum. We agree with the reviewer that this missing piece of evidence would complete the story, but that in its absence the evidence of a lack of mimicry is still compelling.

More generally, it does seem that one might (snarkily) suggest that perhaps ‘much ado is being made about nothing’ given that house crows and common mynas are not ejector species and Asian koel do not evict host eggs, further lowering the cost to the host. Accordingly, why would we expect mimicry to have evolved – there would have been little selection for host-specific mimicry to evolve in these species? That long-tailed strikes do seem to reject model (non-cryptic eggs) and are – at least currently – a frequently-used host is more interesting but again little evidence of mimicry was found. The large size of the Asian koel eggs laid in nests of long-tailed strikes (which have the smallest eggs of the 3 host species) may well be an adaptation to reduce eggs rejection by the hosts, as the authors note, although this is still speculation.

Response: The reviewer raises a good question about whether egg rejection would be predicted, given that two of the hosts rarely reject cuckoo eggs. However, the existing evidence presents an interesting conundrum; house crows do not reject cuckoo eggs, yet the koel egg is described as mimicking that of house crows. This is counter to theoretical expectations that mimicry of host eggs by a cuckoo will evolve in response to egg rejection by the host. A possible explanation could be that there is low rejection by house crows because egg mimicry constrains detection of cuckoo eggs, but Begum et al. 2012 showed that crows and mynas do not even reject non-mimetic eggs, whereas shrike strongly reject non-mimetic model eggs and surprisingly accept koel eggs which are quite different from their own eggs. Further mimicry could also occur in egg size, shape, UV reflectance and pattern, not just color. Thus, we were investigating all aspects of egg appearance to get to the bottom of this conundrum.

The authors raise the possibility (lines 58 to 62) that perhaps “moderate mimicry” might be maintained in generalist parasites such that parasite eggs do not accurately match those of any single host but do so ‘moderately” allowing the parasite at least some opportunities to parasitize a broader variety of hosts. The question for me , then, is – what constitutes ‘moderate’ mimicry? I don’t have an operational sense of that and it seems to be a rather open ended and vague concept. How would one reject that hypothesis?

Response: We have revised this part and decided to remove the term moderate mimicry to avoid any confusion. The term ‘imperfect mimicry’ is most often used in the literature and its definition depends on the parameters of the trait under investigation. In the case of cuckoo egg colour, imperfect mimicry usually refers to an egg colour that is exactly intermediate between the colors of the various host eggs in avian visual space, e.g. Feeney et al. (2014) ‘Jack of all trades’ egg mimicry in the brood parasitic Horsfield’s bronze-cuckoo? Behavioral Ecology, 25: 1365-1373. See line 59-64.

My only other comment is simply that the discussion seems rather long (almost 6 pages) given that there was no evidence of mimicry. I realize the authors are being thorough in discussing a myriad of possible reasons why mimicry has not evolved (e.g., including speculations about the mafia hypothesis, imperfect adaptation, predation rather than parasitism being a driving selection force, or that parasitic Asian koels may remove other koel eggs, etc.). These are all interesting and valid, but quite speculative; perhaps too much discussion is devoted to possible reasons why egg-mimicry does not exist, but for which there is little evidence. The discussion could be shortened by a couple of pages without expensive speculation.

Response: We have reduced the discussion to some extent, focusing more on aspects that we have clear support for in the current results.

In sum, I did not find any major problems with this analysis. The statistics seem to be adequate although I would suggest that perhaps not every koel egg be treated as an independent sample but rather be “nested” within each nest. I realize nest identity was included as a random intercept but I was not sure that would account for non-independence of each koel egg. Again, not a critical issue.

Response: As described on lines 167-169, based on visual inspection, koel eggs within the same host nest were different on appearance, thus, assumed to originate from different female koels. Therefore, clutch identity was only included for host eggs, accounting for non-independence of eggs sharing the same mother.

Overall, I think this manuscript will be a useful contribution, not only to clarify that this species does not exhibit host-specific egg mimicry – as apparently has been claimed – but perhaps even more so as an example of a thorough effort to explore many dimensions of egg mimicry for an obligate brood parasite and several of its host species.

Response: Thank you for acknowledging this thorough approach.

Reviewer #4: This study explores variations in brood parasitic (Asian koel) eggs laid in three different host species. The aim of the study is to determine whether the Asian koel, to some extent, mimics the eggs of its host species. Based on analyses of egg volume, shape, spotting patterns and colour (in the context of avian color vision), the authors found little evidence on the existence of egg mimicry in Asian koel eggs. Furthermore, some characteristics of koel eggs (e.g., egg volume) appear to mismatch the phenotype of host eggs.

Overall, the manuscript is well written and provides quite a lot of details about the study system. The Methods section is clear and provides a complete overview on the general procedures and analyses. The Introduction section is a bit long and I feel that authors go overboard with speculative arguments to explain the lack of egg mimicry in Asian koel eggs. The evolution of mimicry in host-parasite systems is closely related to the costs imposed by parasitism and host rejection abilities. In my opinion, authors should focus their discussion on this point and reduce (not completely eliminate) speculative explanations (eg., tolerance and critical eggs), unless they have evidence that these mechanisms work in their systems. Some of my concerns have been addressed by the authors in their responses to reviewers' comments made to the previous version.

Response: We have revised the manuscript accordingly, reducing the introduction to some extent, and shaping the discussion more to our main focus. 

Other comments:

Line 48: “parasite mimicry”. Please change to “egg mimicry”.

Response: We have changed this to “egg mimicry”. See line 48.

Line 48-51: Although factors such as "climatic variables" may play a relatively important role in certain systems, it should be clearly emphasized that the evolution of egg mimicry is closely related to the evolution of host defenses (i.e., egg rejection).

Response: We have included that the main driver of egg mimicry is host defense, see lines 51-52.

Line 78: “koel chick tolerates…”. Do the authors have evidence of tolerance mechanisms in the Asian koel?

Response: We have revised this. See line 83-84.

Line 70-71: While examination by spectrophotometry often reveals important information on egg mimicry, I wouldn't say that great-spotted cuckoo eggs are "quite different” from magpie eggs (in many cases they may be almost indistinguishable to the human eye). Please soften this statement.

Response: We have revised this. See line 69-72.

Lines 72-76: Any mention to cryptic eggs should be integrated in the paragraph about egg mimicry. I think such a paragraph is a bit out of place in the line of argument of introduction.

Response: We have now integrated this paragraph with the previous paragraph on egg mimicry. Also, we have elaborated in some points of cryptic eggs as Reviewer 2 suggested (see Reviewer 2 point 1). See manuscript lines 72-82.

Line 387: Indeed, there is experimental evidence that eggs volume is an important factor determining egg rejection (Soler at al., Relationships between egg-recognition and egg-ejection in a grasp-ejector species. 2017. Plos One 12(2)).

Response: As suggested by the reviewer, to revise and reduce some of the speculative arguments from the discussion, we have removed this paragraph.

---

## [Editor Report · Decision Letter 1]

17 Jun 2021

No evidence of host-specific egg mimicry in Asian koels

PONE-D-21-04869R1

Dear Dr. Nahid,

We’re pleased to inform you that your manuscript has been judged scientifically suitable for publication and will be formally accepted for publication once it meets all outstanding technical requirements.

Kind regards,

Petr Heneberg

Academic Editor

PLOS ONE

---

## [Editor Report · Acceptance letter]

29 Jun 2021

PONE-D-21-04869R1 

No evidence of host-specific egg mimicry in Asian koels 

Dear Dr. Nahid:

I'm pleased to inform you that your manuscript has been deemed suitable for publication in PLOS ONE. Congratulations! Your manuscript is now with our production department. 

Kind regards, 

on behalf of

Dr. Petr Heneberg 

Academic Editor

PLOS ONE